# Egocentric Vehicle Dense Video Captioning

## ABSTRACT

Typical dense video captioning mostly concentrates on third-person videos, which are generally characterized by relatively delineated steps among events as seen in edited instructional videos. However, such videos do not genuinely reflect the way we perceive our real lives. Instead, we observe the world from an egocentric viewpoint and witness only continuous unedited footage. To facilitate further research, we introduce a new task, Egocentric Vehicle Dense Video Captioning, in classic first-person driving scenario. This is a multi-modal, multi-task project for a comprehensive understanding of untrimmed, egocentric driving videos. It consists of three sub-tasks that focus on event location, event captioning, and vehicle state estimation separately. For the purpose of accomplishing these tasks, it is necessary to deal with at least three challenges, those are extracting relevant ego-motion information, describing driving behavior and understanding the underlying rationale, as well as resolving the boundary ambiguity problem. In response, we devise corresponding solutions, encompassing a vehicle ego-motion learning strategy and a novel adjacent contrastive learning strategy, which effectively address the aforementioned issues to a certain extent. We validate our method by conducting extensive experiments on the BDD-X dataset, all of which show promising results and achieve new state-of-the-art performance on most metrics, which proves the effectiveness of our approach.

## CCS CONCEPTS

• **Computing methodologies → Video summarization**.

## KEYWORDS

Egocentric Vehicle Video, Video captioning, Contrastive Learning, Ego-motion

**ACM Reference Format:**

Anonymous Author(s). 2018. Egocentric Vehicle Dense Video Captioning. In *Proceedings of Make sure to enter the correct conference title from your rights confirmation emai (Conference acronym 'XX)*. ACM, New York, NY, USA, 10 pages. https://doi.org/XXXXXXX.XXXXXXX

## 1 INTRODUCTION

Dense Video Caption (DVC) is a branch of video understanding that aims to locate and describe all events within an untrimmed video[30, 40, 56, 66, 70, 74]. Leveraging DVC allows for more efficient video processing, like detailed content retrieval and intelligent surveillance. Several related research has been conducted in some

specific scenarios, such as procedural human activities[4] and instructional videos[58, 73], all these efforts have seen remarkable capabilities.

However, the majority of DVC researchers focus on analyzing exocentric, or third-person videos, yet investigations into an egocentric, or first-person, viewpoint remain rare. Different from the former, which commonly narrates details about other objects within the camera lens as shown in Figure 1(b), this view primarily concerns the movements of camera wearer [8, 17, 26, 34, 51] itself. In fact, it is the egocentric perspective that precisely reflects the authentic and natural manner in which human beings and autonomous agents observe their surroundings[8, 17, 35, 60, 61]. Besides, it fundamentally affects how we understand and engage with our environment on a daily basis, by influencing our perception[39, 49], decisions[22, 46, 69] and interactions[36, 41, 59] in the complexities of the world around us[54, 69]. Consequently, this gives rise to Egocentric Dense Video Captioning (Ego-DVC). In contrast, Ego-DVC is characterized by its unique competence to learn motion changes of the view from untrimmed egocentric videos, which corresponds closely to real-life experiences and is thus more practical and meaningful. There are plenty of potential applications for this subject, among which, driving is a classic domain.

For the facilitation of investigation, we present a new task, Ego Vehicle Dense Video Caption. It is devoted to driving scenarios, where a camera is mounted at a specific location on the vehicle, to capture multiple variable ego-motion information and the evolving landscape as the vehicle moves. Ideally, the information recorded should implicitly include the locations of various driving behaviors, corresponding descriptions and rationales, as well as vehicle states, hence we propose such a task. As shown in Figure 1(a), given a sequence of first-person driving frames, we are tasked with three sub-tasks. a) Event Location, aiming to identify all driving events that take place in the video, while also simultaneously pinpointing the precise start and end timestamps for each. b) Caption generation, intending to describe the actions of all detected events in natural language and provide their contextually relevant rationale. c) Vehicle State Estimation(VSE), attempting to estimate specific vehicle states for the whole video, including velocity and steer, these states describe the fundamental motion patterns of the vehicle, for example, we might infer a vehicle is slowing down if we detect that its velocity values are consistently decreasing during a period of time.

Upon a profound analysis, we suppose our task is confronted with at least three distinct challenges:

**Ego-Motion Information**: For egocentric videos, ego-motion information is embedded in the dynamic changes of the camera lens. However, in the complex road environment, irrelevant objects and deterministic signals[62] for driving are always intricately intertwined and subtly changing, making it difficult to extract effective motion representation.

**Description and Rationale**: Analyzing the cause is often more difficult than describing the problem, this also applies to our task.

**Figure 1: A schematic illustration of Ego Vehicle DVC. Compared with typical DVC, Ego Vehicle DVC comprises four outputs: location, description, as well as additional rationale and vehicle state.**

As shown in 1(b), the reason the car slows down is that it encountered a red light, which is not easy to analyze because the accurate rationale(traffic light) that truly corresponds to the description in driving scenarios often appears trivial and is prone to be overlooked, it requires a thoroughly probe into the videos.

**Boundaries Ambiguity**: Compared to typical DVC, the egocentric videos are always continuous, unedited real-life footage. Additionally, those captured by vehicle-mounted cameras predominantly feature the road, with the majority of the view occupied by it and only several impalpable changes occurring, especially in suburbs. As shown in Figure 1(b), frames near the event boundary are very similar thus causing ambiguous, yet traditional DVC in Figure 1(a) generally does not encounter this issue. This feature obstructs the precise identification of the boundary between two adjacent driving behavior events.

To address these challenges, we intentionally designed corresponding strategies. Concerning the first two challenges, we introduce a vehicle ego-motion learning strategy, it integrates a pre-trained extractor and VSE module, the former function on extracting ego-motion features incorporating driving-decision, and the VSE module is applied to strengthening the representation with specific motion values(vehicle state). Regarding the third one, we devise an adjacent contrastive learning(ACL) strategy that enhances event representation by performing contrastive learning among the three modalities of adjacent events. This approach is capable of reducing the ambiguity of event boundaries and thereby distinguishing them more clearly.

To summarize, our main contributions are three-fold:

(1) We introduce the Ego Vehicle DVC task, allowing for a detailed multimodal comprehension of untrimmed egocentric driving videos.

(2) We pioneered incorporating an ego-motion information learning strategy in DVC. Besides, we design an adjacent contrastive learning strategy for event representation learning.

(3) We conducted extensive experiments on the BDD-X dataset, achieving state-of-the-art results in most metrics, thereby demonstrating the effectiveness of our approach.

## 2 RELATED WORK

### 2.1 Egocenertic Vision

Egocentric vision, providing a distinctive and intuitive perspective on human interactions with the environment[8, 35, 36, 41, 59, 61], is thriving increasingly. In contrast to traditional tasks, which typically process well-defined exocentric videos curated by photographers, egocentric videos possess their unique features that remain underexplored, such as view changes[20], even the currently popular large language model (LLM) still performs poorly on this issue[8]. To delve deeper, a wide range of related topics are gradually emerging and attracting the attention of researchers.

Egocentric human-object interaction(EGO-HOI) is a vital task in this field, it primarily concentrates on the interactions between hands and objects from an egocentric viewpoint[9, 17, 53, 64]. Some research attaches importance to hand pose estimation and object-centric representations[1, 64], others strive to learn reasoning and indirect reference through question-answering on real-world egocentric footage[25, 26], and further works on captioning egocentric videos by cross-view transfer learning from exocentric sources[21, 63]. EGO-HOI paves the way for nuanced communication between humans and external entities, while it primarily focuses on specific targets, lacking a comprehensive understanding of the overall scenario.

Egocentric Visual Perception, which generally serves as the eye of the entire autonomous system, is a crucial part of this topic. It is widely applied in applications such as Virtual Reality (VR) and Augmented Reality (AR), where a fundamental task involves locating the 3D positions of multiviewed visual queries in complex scenarios [17, 37]. Embodied AI, a prevalent topic at present, is also inextricably linked to this technology. [13, 54, 72] endeavors to master comprehensive 3D scene understanding skills, enabling real-world embodied agents to execute commands effectively. Autonomous driving is another typical application of egocentric visual perception, studies such as[19, 38, 47]attempt to plan vehicle action based on surroundings and achieve impressive outcomes, despite the understanding of the underlying rationale remains elusive. Furthermore, Several works consider detecting regions pertinent to driving decisions[12, 28, 45], although these methods are useful for

those familiar with traffic laws, they may confuse ordinary users. Therefore,[27, 29, 65, 68] strive to comprehend vehicle motion with rational and straightforward natural language, while they limit to a short-term video with one main behavior which is not consistent with reality.

## 2.2 Dense Video Captioning

Dense video captioning (DVC) is a multi-task project that requires identifying events and generating captions for them. Originally, this task predominantly employed two-stage methodologies [23, 24, 30, 52, 57, 70], starting with the temporal localization of events, followed by their captioning. This paradigm heavily depends on the performance of the first stage, consequently, many have begun to consider the end-to-end one-stage approach [6, 7, 10, 31, 40, 44, 48, 55, 56, 66], intending to reach mutual improvement by coordinating the interaction of jointly training two sub-tasks. Learning from Bert[11], [70, 74] design mask mechanisms for the interaction between the two modules separately. The emergence of DETR[5] brought fresh prospects to the task, based on which Deformable-DETR[75], [56] employs a set prediction scheme to elegantly parallelize the two sub-tasks. Building on this foundation, [55] introduces contrastive learning to enhance event representation through the contrast between events and captions. Given the information embedded in the audio tracks of these videos,[23, 24, 66] take a unique view by extracting the audio features from the videos, thereby significantly improving performance.

However, the methods designed for edited and exocentric videos fall short of meeting our task's requirements, which focused exclusively on analyzing unedited, audio-free, continuous, real-world driving footage from a first-person perspective. In Driving scenarios, the views are typically dominated by the road surface, and present minimal variation in the surrounding environment, leading to ambiguous distinctions between events and complicating event localization. What's more, the inherent nature of egocentric videos concerning dynamic view changes, adding another layer of complexity to video analysis.

## 3 METHODOLOGY

In this work, we focus on locating all driving behavioral events, captioning behavior descriptions and rationales, as well as estimating vehicle state values throughout the entire ego-vehicle video. Figure 2 provides a graphical illustration of our comprehensive framework. Initially, we pre-train a vehicle ego-motion extractor(Sec 3.1), leveraging which frame features will be captured and fed into a DETR-based architecture and be further amplified by VSE module (Sec 3.1) at the end of the encoder, the decoder will generate several event representations, we then introduce a novel adjacent contrastive learning (Sec 3.2) strategy to enhance semantic representation of these event and finally generate all descriptions and rationales.

## 3.1 Vehicle Ego-motion Learning Strategy

In this section, our objective is to introduce the vehicle ego-motion learning strategy. It comprises two parts, the first part involves pre-training a vehicle ego-motion extractor by self-supervised learning,

while the second part enhances ego-motion representation through supervised VSE.

**Vehicle Ego-motion Extractor**.

In this part, we try to achieve an extractor that allows us to map the raw frame input to a compact representation containing basic ego-motion information, which is essential for our three sub-tasks. Considering that under normal circumstances, driving decisions should be consistent with actual behavior, we attempt to incorporate driving-decision awareness, expecting to assist in extracting appropriate motion representation while also taking into account the crucial visual cues based on the current scenario. Following the design of PPGeo[62], our pre-training progress consists of two stages.

**Self-supervised photometric reconstruction.** Photometric Reconstruction aims to reconstruct the scenario by learning photometric differences, or more specifically, standard color constancy between frames. There exists a prevalent method that enables the model to translate input pixels into ego-motion and detailed scene architecture as well as estimating camera intrinsics.

We follow [16, 71] to perform photometric reconstruction by jointly training a PoseNet and DepthNet across two frames. PoseNet is designed to estimate the 6-DoF ego-motion and camera intrinsics between consecutive frames, and DepthNet predicts the depth map in the meantime. We employ a ResNet and MPViT to serve them separately. Assuming that we want to reconstruct the t-th frame $I_t$ from $I_{t-1}$. the pixel-wise color constancy can be reconstructed as follows:

$$I_{t'} = I_{t-1} \langle \text{proj} (D_{t-1}, T_{t-1 \to t}, K) \rangle \tag{1}$$

here $I_{t'}$ is the reconstruction of frame $t$, proj() indicates the operation with which we project original pixels space of $I_{t-1}$ into predicted 2D coordinates making use of depth map $D_{t-1}$ from and relative pose $T_{t-1 \to t}$ between $I_{t-1}$ and $I_t$. Afterward, utilizing bilinear interpolation, we sample values to create $I_{t'}$ through <> operation. As for camera intrinsics $K$, we consider it a constant value and assess it by calculating the average of $K_{t-1}$ and $K_t$ predicted from relevant frames.

Align with [16, 71], We calculate the loss according to the following formula:

$$\mathcal{L} = \lambda_{pe} \mathcal{L}_{pe} + \lambda_s \mathcal{L}_s \tag{2}$$

here $\mathcal{L}_{pe}$ represents photometric loss comprised of structural similarity index measure(SSIM) and $L_1$ term:

$$\mathcal{L}_{pe} = \frac{\alpha}{2} (1 - \text{SSIM} (I_t, I_{t'})) + (1 - \alpha) |I_t - I_{t'}| \tag{3}$$

$\mathcal{L}_s$ represent disparity smooth-ness loss:

$$\mathcal{L}_s = \left|\partial_x d_t^*\right| e^{-|\partial_x I_t|} + \left|\partial_y d_t^*\right| e^{-|\partial_y I_t|} \tag{4}$$

where $d_t^*$ is the mean-normalized inverse depth map.

**Vehicle ego-motion extractor**. After the preceding phase, we will obtain a DepthNet and a PoseNet. This PoseNet can capture relative motion differences between two adjacent frames. In fact, what we really need is the "difference" on a certain frame, it essentially means learning the driving policy, that is to say, performing suitable driving behavior based on current observation. To bridge this gap, we follow the methodology outlined in [62], wherein $I_t$ is removed and retain only one $I_{t-1}$ as input. In addition, we freeze

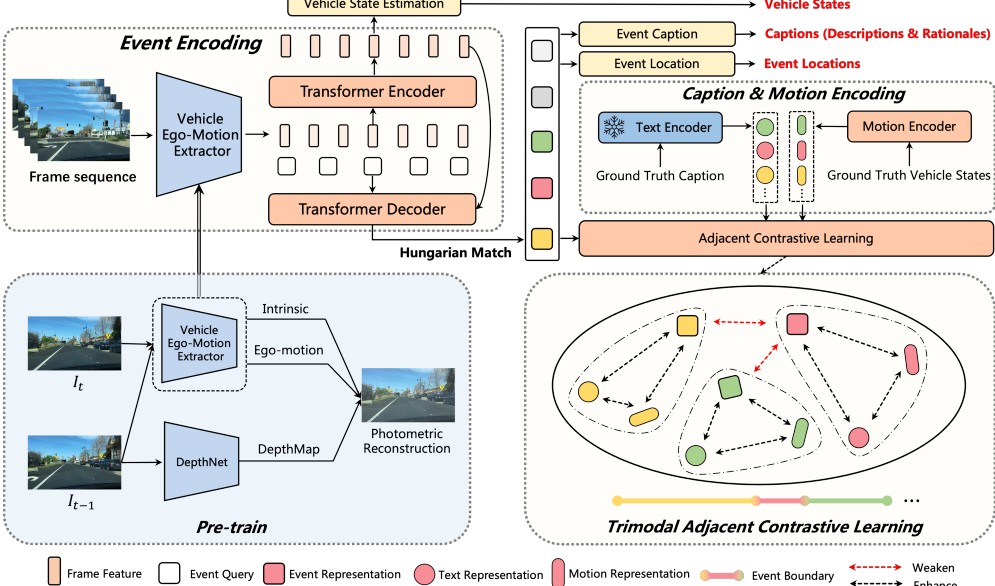

**Figure 2: A overview of our framework. The vehicle ego-motion extractor is pre-trained in advance through photometric reconstruction. Event Encoding is responsible for encoding frame sequences into vehicle motion representation for state estimation as well as event representations for location and captioning. Ground truth captions(descriptions and rationales) and motion values(vehicle states) are encoded in the Caption & Motion Encoding module by the corresponding encoder independently. In Adjacent Contrastive Learning, these three modalities enhance each other within the same event(shown in the same color), while simultaneously weakening their adjacent events(yellow and pink), but ignoring interaction with temporally distant events(yellow and green).**

the DepthNet, reinitialize the parameters of PoseNet, and conduct retraining of the entire model as the preceding phase once again. Ultimately, a new PoseNet will be obtained with the assistance of the well-trained DepthNet, referred to as the vehicle ego-motion extractor. It enables the acquisition of not only ego-motion representations but also critical decision-making information, that is, the rationale.

**Vehicle State Estimation**.

In this part, we intend to introduce VSE, a special module that plays a dual role within our framework. The major function is to carry out the task of vehicle state estimation. Simultaneously, since the vehicle state is, in essence, a concrete form of ego-motion, VSE and vehicle ego-motion representation can be mutually beneficial throughout this supervised learning process.

Assuming that an ego-vehicle video $V$, consisting of $N$ frames, is labeled as $\{(v_1, s_1), (v_2, s_2), ..., (v_{M-1}, s_{M-1}), (v_M, )\}$, here $v_t$ denotes the velocity at the $t$-th timestamp, $s_t$ denotes the steering value between the $t$-th and $(t + 1)$-th records. It is important to notice that the collected vehicle states do not always align with frame sequences. In practice, they are recorded at fixed intervals, rather than frame by frame, therefore $M$ is usually less than $N$. We will feed into $N$ frames and extract features through the vehicle ego-motion extractor, after transformer encoder, the features $f_1, f_2...f_N$ will be applied to estimate velocity and steer of the ego-vehicle throughout the duration as follows:

$$f_1, f_2...f_N = \text{BiLSTM}(f_1, f_2...f_N) \tag{5}$$

Given $M$ records associated with $N$ frames, we utilize linear interpolation to downsample $N$ features to $M$. Note that we use multi-scale features with CNN, to accumulate them we apply max pooling results in $f_1, f_2...f_M$, then we map them into scalars as follows by MLP:

$$v'_1, v'_2...v'_M = \text{MLP}_v(f_1, f_2...f_M) \tag{6}$$

$$s'_1, s'_2...s'_{M-1} = \text{MLP}_s(f_1, f_2...f_{M-1}) \tag{7}$$

Here $v'_i$ represents estimated velocity, $s'_i$ represents steer between $f_i$ and $f_{i+1}$.

Finally, We calculate the loss using Mean Square Error:

$$\mathcal{L}_{mse} = \frac{1}{M} \sum_{i=1}^{M} (v_i - v'_i)^2 + \frac{1}{M-1} \sum_{i=1}^{M-1} (s_i - s'_i)^2 \tag{8}$$

## 3.2 Adjacent Contrastive Learning Strategy

In this section, we will introduce our contrastive learning strategy particularly devised for driving events location. The core concept involves applying three types of modalities to adjacent driving behavioral events. In this task, we regard an untrimmed ego vehicle video $V$ as a set $E = \{e_n | e_n = (l_n, c_n), n = 1, 2, ..., N\}$, where $l_n = (ls_n, le_n)$ defines the time location for event $n$, starting at time $ls_n$ and ending at time $le_n$, $c_n = (d_n, r_n)$ provides the caption for event $n$, with $d_n$ offering description of driving behavior, $r_n$ offering its

rationale. For convenience, We define a new set $E_n$ consisting of event $n$ and its adjacent events as follows:

$$E_n = \{e_i \in E | (le_i \geq ls_n \wedge ls_i \leq le_n) \vee \\ (|le_i - ls_n| \leq \varepsilon \vee |ls_i - le_n| \leq \varepsilon)\} \tag{9}$$

where $\varepsilon$ is a constant, defining the temporal distance threshold of adjacent events.

As seen in Figure 2, the representations of frame sequences flow into the Event Encoding module and are encoded into $M(M \leq N)$ predicted driving behavioral events. Sequencely, after a typical Hungarian Match, they will be matched with ground truth. To deepen discriminability, we introduce an additional head for event representation learning. This enables us to project the predicted events into $M$ semantically representations $Ep = \{ep_1, ep_2, \ldots, ep_M\}$. Ideally, we consider they should satisfy at least the following three criteria:

- They should fully encapsulate the vehicle ego-motion information of the corresponding behavioral events.
- Any two adjacent behavioral events should be clearly distinguished.
- Any two behavioral events that are distant in time should not affect each other.

Taking these three considerations into account, we devise a novel contrastive learning method. Previous methods usually calculate the loss in a global range, even extending to a training batch[18, 43] attribute to the variety of their data pairs. On the contrary, we merely confine ourselves to an adjacent range. This is because the scope of behaviors observed in ego-vehicle videos is generally limited and prone to repetition. A vehicle described as "The car turns right at an intersection" during a certain period may likely be described by the same sentence again before long, this phenomenon is so common in vehicle scenarios that previously prevalent contrast learning would result in a significant decline in effectiveness. However, we can make sure that the driving behaviors of the neighboring events certainly contain obvious differences, otherwise they would not be divided into two events. Our strategy consists of the following three parts.

**Event-Caption Contrastive Learning**

The information in egocentric driving videos is not confined to the frame sequences alone, it exists in the associated caption as well. We posit that the essential motion insights from both mediums ought to be consistent. With the prior knowledge of relevant textual features, we can enhance the semantics of events and achieve cross-modal alignment between video and caption content. Clip[43] is a standard work related to this idea. Unfortunately, as a generation task, it is impossible to access caption representation before generating it, which leads to a deadlock situation.

To break this deadlock, we introduced a pre-trained text encoder to encode all captions($d_n$ or $r_n$) into $C$. After aligning, we project $Ep$ and $C$ into a shared space and calculate the cross-modal cosine similarity matrix between the projected embeddings as $\omega^{ec} \in \mathbb{R}^{M \times N}$, we will calculate our adjacent event-caption contrastive loss as follows:

$$\mathcal{L}_{ec} = -\sum_{n=1}^{N} \log \frac{\exp(\omega^{ec}(\text{match}(n), n)/\tau)}{Z_n^{ec}} \tag{10}$$

here $Z_n^e$ is a modified normalization factor:

$$Z_n^{ec} = \sum_{i=1}^{M} \begin{cases} \exp(\omega^{ec}(\text{match}(i), n)/\tau) & \text{if } e_i \in E_n \\ 0 & \text{else} \end{cases} \tag{11}$$

where match() devotes the matching operate from $E$ to $Ep$, $\tau$ signifies a temperature ratio.

**Event-Motion Contrastive Learning**

Fundamentally, the generation of driving videos is attributed to the driver issuing signals for steering and velocity based on a certain observation, therefore, the information entailed in the vehicle states should also maintain consistency with the behavioral events. As a result, we further implemented cross-modal semantic alignment between vehicle motion and event. This approach enhances the semantic representation of events by incorporating readily captured motion information.

Just as in Event-Caption Contrastive Learning, we adopt a motion encoder to encode the ground truth vehicle motion state values of current events into a representation resulting $S$, After aligning and projecting into a joint space, we calculate the cross-modal cosine similarity between $Ep$ as $S$ as matrix $\omega^{es} \in \mathbb{R}^{M \times N}$. Same as before, contrastive loss $\mathcal{L}_{es}$ will be derived in the same manner.

**Motion-Caption Contrastive Learning**

We present the last contrastive Learning between motion and caption, aiming for a mutual complementarity of information between the two modalities, and further indirectly enhancing event representation.

Differing slightly from the former two components, the score matrix will result in a square matrix $\omega^{mc} \in \mathbb{R}^{N \times N}$, the motion-caption contrastive loss is determined using the following formula:

$$\mathcal{L}_{mc1} = -\sum_{n=1}^{N} \log \frac{\exp(\omega^{mc}(n, n)/\tau)}{Z_n^{mc1}} \tag{12}$$

where $Z_n^{mc1}$ calculate the normalization factor vertically of $\omega^{mc}$, we can also acquire $\mathcal{L}_{mc2}$ with another $Z_n^{mc2}$ normalized horizontally, then the loss is expressed as follows:

$$\mathcal{L}_{mc} = \frac{1}{2}(\mathcal{L}_{mc1} + \mathcal{L}_{mc2}) \tag{13}$$

Finally, our complete adjacent contrastive learning loss $\mathcal{L}_{cl}$ is signified as following expression:

$$\mathcal{L}_{cl} = \alpha \mathcal{L}_{ec} + \beta \mathcal{L}_{em} + \gamma \mathcal{L}_{mc} \tag{14}$$

Where $\alpha + \beta + \gamma = 1$, and $\alpha, \beta, \gamma$ are three trainable paramaters.

Principally, as the Trimodal Adjacent Contrastive Learning module depicted in Figure 2, we utilize the three modalities within a certain event to achieve cyclical contrastive learning, thereby enabling mutual enhancement among them and meeting the first point of the criteria we mentioned earlier. Furthermore, our method, by limiting the scope of contrast, not only weakens the semantic relevance of adjacent events but also concurrently avoids the influence of temporally distant events, satisfying the second and third points of the criteria, thus ultimately achieving the purpose of distinctly distinguishing the representation of adjacent events.

## 4 EXPERIMENTS

**Dataset**. To the best of our knowledge, there are no datasets perfectly aligned with the requirements of our task, we attempt to

**Table 1: Comparison to the state of the art for event location, w/o vs indicates discarding vehicle states**

| Method | Feature | Recall | | | | | Precision | | | | | F1 |
|---|---|---|---|---|---|---|---|---|---|---|---|---|
| | | 0.3 | 0.5 | 0.7 | 0.9 | agv | 0.3 | 0.5 | 0.7 | 0.9 | agv | |
| MT[74] | R34 | **84.78** | **70.01** | **58.18** | 28.64 | **60.42** | 86.69 | 57.14 | 27.26 | 4.96 | 44.01 | 50.93 |
| ESGN[40] | R34 | 75.16 | 50.78 | 26.26 | 11.34 | 40.89 | 91.58 | 58.97 | 29.31 | 11.37 | 47.81 | 44.08 |
| UEDVC[70] | R34 | 84.52 | 59.22 | 39.27 | 18.17 | 50.30 | 88.17 | 62.07 | 38.51 | 14.40 | 50.78 | 50.54 |
| GVL[55] | R34 | 77.06 | 58.30 | 41.27 | 17.81 | 48.61 | 91.02 | 64.84 | 38.86 | 16.90 | 52.90 | 50.67 |
| PDVC [56] | R34 | 72.91 | 57.61 | 45.08 | 23.57 | 49.79 | 90.73 | 67.62 | 44.18 | 19.63 | 55.54 | 52.84 |
| Ours (w/o VS) | $VEM_{r34}$ | 73.37 | 59.26 | 45.32 | 22.89 | 50.21 | 91.47 | 74.15 | 52.86 | 21.31 | 59.95 | 54.65 |
| Ours | $VEM_{r34}$ | 72.99 | 59.02 | 45.61 | 25.52 | 50.79 | **94.97** | **77.02** | 54.29 | **25.92** | **63.05** | **56.26** |

**Table 2: Comparison to the state of the art for captioning, w/o vs indicates discarding vehicle states**

| Method | Feature | Description | | | | | Rationale | | | | |
|---|---|---|---|---|---|---|---|---|---|---|---|
| | | B4 | M | R | C | S | B4 | M | R | C | S |
| MT[74] | R34 | 8.58 | 12.51 | 24.92 | 49.20 | 4.06 | 2.89 | 6.42 | 13.70 | 29.09 | 2.56 |
| UEDVC[70] | R34 | 18.77 | 19.22 | 33.16 | 131.68 | 22.64 | 3.08 | 10.51 | 15.10 | 36.05 | 10.79 |
| GVL [55] | R34 | 18.01 | 21.39 | 36.76 | 140.89 | 25.63 | 3.80 | 11.04 | 17.35 | 36.32 | 11.45 |
| PDVC[56] | R34 | 18.68 | 22.23 | 37.70 | 141.90 | 24.21 | 4.37 | 10.12 | 18.64 | 49.61 | 10.55 |
| Ours (w/o vs) | $VEM_{r34}$ | 20.42 | 25.43 | 41.06 | 153.94 | 23.95 | 5.11 | 10.92 | 19.69 | 54.80 | 12.13 |
| Ours | $VEM_{r34}$ | **21.82** | **25.42** | **42.47** | **162.12** | **26.37** | **5.62** | **11.25** | **21.60** | **59.79** | **12.47** |

evaluate our proposed approach on the BDD-X[29], a widely used ego-vehicle video dataset derived from BDD100K[67] for short-term captioning which encompasses over 77 hours of driving within 6984 videos. Every video lasts about 40 seconds on average and comprises approximately 1 to 5 driving behavior events and their location, each annotated with a description and rationale. However, This dataset initially doesn't consider vehicle state, we have to acquire it by mapping video id to the original BDD100K. Due to version changes, there exist only 4641 corresponding videos, with 3578 for training, 524 for validation, and 539 for testing. In BDD-X, the GPS information is collected at 1Hz using the same equipment, suggesting the camera intrinsics are identical, that's the reason we can estimate a single group of this parameter during the self-supervised photometric reconstruction stage at Sec 3.1. Velocities are recorded directly by the GPS, while for steering, we apply the course message (angle relative to geographic true north) as pseudo-values between two consecutive GPS records.

**Implementation Details**. To pre-train the vehicle ego-motion extractor, we use ResNet34 and MPViT as the PoseNet and DepthNet respectively. For each stage, it takes about 5 days on 8 Tesla V100 GPUs to train for 20 epochs with batch sizes of 32 and 64.

During training, we employ a frozen Roberta model as the text encoder and a BiLSTM model for the vehicle states encoder; Following PDVC[56], our method is based on a deformable-DETR with two encoder-decoder layers of 512 dimensions and uses LSTM-DSA [56] serve as event caption head to generate captions; Contrastive learning is used only in training and ignored at the inference stage, we set the temperature $\tau$ to 0.1, temporal threshold $\varepsilon$ to 3; All events will be sorted by their confidence scores, and the number of predictedk events will be decide according to a CounterHead. We

set the batch size to 2 and trained for 30 epochs in a Tesla V100 using Adam with a learning rate of 0.0001 and a weight decay of 0.0001.

**Evaluation metrics**. As a multi-task model, We evaluate our method in three aspects: 1) For VSE, we employ root mean squared error (RMSE) and threshold accuracies $A_\tau$. It calculates the ratio of test samples that have prediction errors smaller than a predefined threshold $\tau$, which we set at multiple levels: {0.1, 0.5, 1.0, 5.0}. 2) For events location, We calculate the average precision and average recall for IoU thresholds set at {0.3, 0.5, 0.7, 0.9} and their harmonic mean, the F1 score. 3) For captioning, we follow the widely utilized evaluation tool provided by ActivityNet Challenge 2018[15] adopting BLEU4(B4)[42], METEOR(M)[2], ROUGE_L(R)[33] and CIDEr(C)[50] to measure matched pairs between generated caption and ground truth across IOU thresholds of {0.3, 0.5, 0.7, 0.9}. Taking into the quality of the story of the whole drive video, we additionally use SODA_c(S)[14] for an overall evaluation.

## 4.1 Comparison with State-of-the-art Methods

Since there is currently no task that aligns completely with ours, we primarily compare event location and dense caption with current DVC tasks. We compare five approaches on the BDD-X dataset using their official codebases. MT[74] is the first one to utilize a transformer in this field, we pick up the top 30 events for indicator calculation; ESGN[40] offers a reinforcement learning approach, however, a proposal method is needed to extract candidates in advance, we adopt ActionDetection-DBG[32] instead of its original SST[3] and select top 100, noting that the official codebase solely contains the event sequence generation stage, we only compare in this subtask. UEDVC[70] transforms event-location into a sequence

generation problem and proposes three pre-training tasks to effectively reinforce the correlation between sub-tasks. GVL[55] is slightly similar to our method, utilizing global contrastive learning to strengthen event representation, besides, it utilizes a semantic-aware label assignment mechanism to improve recall. PDVC[56] initially introduces detr into this task, presenting an elegant end-to-end approach. There are also several remarkable works [23, 24, 66] focus principally on audio features, which are not included in BDD-X, so we do not consider comparing with them. For the sake of a fair comparison, we all use the ResNet34(R34) structure as the basic feature extractor, we should keep in mind the vehicle ego-motion extractor(VEM) is also from R34. In addition, we omit vehicle states(vs) and keep only captions, which means discarding the vehicle state estimation task and motion-related contrastive learning module.

**Event location performance**. Table 1 exhibits the effectiveness of our approach and other several state-of-the-art methods. It indicates that we achieve the best results in terms of precision and F1 score, with respect to recall, MT receives the highest score, the reason lying in its inherent emphasis on recall over precision. We witness that although both are from PDVC and applying contrastive learning, the precision of GVL is lower than PDVC in the BDD-X dataset, while the situation is reversed in common datasets such as ActivityNet[4] and [73]. The reason lies in that the modules designed in GVL are better suited for one-to-one style datasets, where events in the video directly match the caption. For datasets like BDD-X, where there is a one-to-many relationship and multiple driving events correspond to the same caption, this approach is not applicable, whereas our method overcomes this defect.

**Dense caption performance**. Table 2 shows the impact on caption generation, namely, description and rationale generation in our task, compared with several previous works. we can observe that our method sets new state of the art on all metrics. From an overall perspective, the effectiveness of rationale generation is much lower than that of description generation, which aligns with our speculation that analyzing the reasons for a phenomenon is much more challenging than describing it.

## 4.2 Ablation Studies

**Adjacet Contrastive Learning**. We conducted extensive experiments to assess the impact of our ACL approach. As illustrated in Table 3, which shows its influence on driving behavior event location. Ground truth vehicle state and caption(description and rationale) are two optional types of potential impact factors of this approach, in addition to the essential event features modality. We also contrast two ways of integrating losses across different modality pairs, direct addition and learnable weighted combination. Although it does not significantly impact the recall rate compared to the approach without ACL, it does substantially improve precision by around 10.33%, resulting in an increase of around 5.01% in F1. This is because our method, designed among adjacent events, can enhance the boundary awareness capability of the model, making it more distinguishable from neighboring events.

To verify the necessity of contrastive learning within the vicinity, we conducted additional experiments to confirm the impact of a temporal distance $\varepsilon$. As depicted in Figure 3, we adjusted $\varepsilon$ from 1s to 40s, with 40s representing engagement in comparative learning

over the entire range of the video. The precision observed in the line graph shows an initial rise followed by a subsequent decline. This pattern emerges because, in driving scenarios, we can only ascertain differences between adjacent driving events; However, it remains uncertain whether two temporally distant driving behaviors are distinct. Therefore, a large $\varepsilon$ may impair results by contrastive learning method.

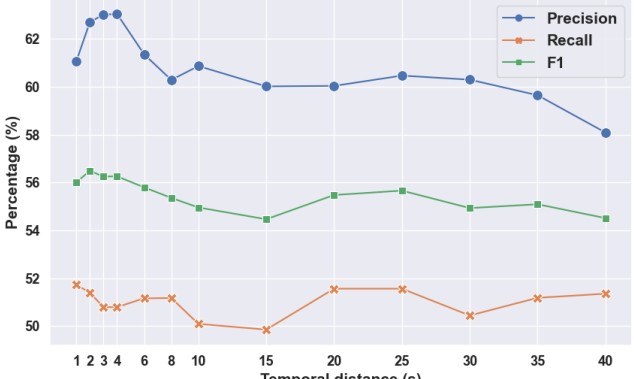

**Figure 3: The event location performance on different temporal distance($\varepsilon$).**

Theoretically, although ACL does not affect caption generation, it can indirectly influence it by affecting event location. Given this possibility, we continue to carry out experiments on the change of text generation capability. Table 4 demonstrates our deduction, we can see our method achieves the best scores in most indicators.

**Vehicle Ego-motion Learning**. We conducted experiments to investigate the impact of different feature extractors on the VSE task. In Table 5, we compared ResNet34 (pre-trained on ImageNet 1k), CLIP (pre-trained on 400 million image-text pairs), and I3D (pre-trained on Kinetics 400) with our VEM extractor trained from on an initialized ResNet34 ($VEM_{r34}$). Among these, ResNet34, CLIP, and $VEM_{r34}$ were applied to image frames, while I3D is a video feature extractor. As shown in Table 5, our $VEM_{r34}$ features performed significantly better than the other two image encoders. The RMSE of steer and velocity decreased by 6.90% and 28.57% respectively compared with the original Resnet34. Moreover, we achieve results comparable to those of the video encoder I3D, even though I3D outperforms ours when the threshold is relatively low. This overall result demonstrates the effectiveness of our method in capturing vehicle ego-motion representation.

Table 6 illustrates the effect of vehicle ego-motion learning strategy, namely extractors and VSE module, on event location and caption. A driving event should contain a trend-oriented behavior, which is closely related to ego motion. Our vehicle ego-motion feature is not only informative regarding ego-motion but also contains driving decision-making, in other words, vehicle behavior rationale information, which exactly meets the requirement. In addition, as a component of our vehicle ego-motion learning strategy, the VSE task is also beneficial to event location and text generation. In table 6, the average of event location with $VEM_{r34}$ almost surpasses all others, especially its counterpart, Resnet34, achieving 3.91%,6.96%,

**Table 3: Ablation of adjacent contrastive learning's effect on event location.**

| Vehicle State | Caption | Method | Recall | | | | | Precision | | | | | F1 |
|---|---|---|---|---|---|---|---|---|---|---|---|---|---|
| | | | 0.3 | 0.5 | 0.7 | 0.9 | avg | 0.3 | 0.5 | 0.7 | 0.9 | avg | |
| × | × | | 72.34 | 61.05 | 44.75 | 23.60 | 50.44 | 90.82 | 67.56 | 50.32 | 19.88 | 57.15 | 53.58 |
| √ | × | | 73.00 | **61.21** | 44.55 | 23.02 | 50.45 | 92.73 | 72.26 | 50.62 | 23.09 | 59.68 | 54.67 |
| × | √ | | 72.58 | 61.03 | 44.24 | 21.08 | 49.73 | 94.19 | 73.65 | 51.18 | 22.11 | 60.28 | 54.50 |
| √ | √ | Add | **73.88** | 60.21 | 45.12 | 23.64 | 50.96 | **95.22** | 76.99 | 53.11 | 25.15 | 62.61 | 56.19 |
| √ | √ | Weighted | 72.99 | 59.02 | **45.61** | **25.52** | 50.79 | 94.97 | **77.02** | **54.29** | **25.92** | **63.05** | **56.26** |

**Table 4: Ablation of adjacent contrastive learning's effect on caption generation**

| Vehicle State | Caption | Method | Description | | | | | Rationale | | | | |
|---|---|---|---|---|---|---|---|---|---|---|---|---|
| | | | B4 | M | R | C | S | B4 | M | R | C | S |
| × | × | | 21.08 | 23.84 | 40.22 | 156.13 | 24.98 | 4.99 | 10.35 | 19.97 | 56.54 | 11.63 |
| √ | × | | 19.96 | 23.92 | 40.13 | 159.25 | 24.12 | 4.81 | 10.56 | 19.44 | 57.38 | 11.02 |
| × | √ | | 20.05 | **25.84** | 40.10 | 156.91 | 24.55 | 5.02 | **11.25** | 19.89 | 55.13 | 11.36 |
| √ | √ | Add | 21.02 | **25.84** | 41.05 | 160.25 | **26.38** | **5.64** | 10.82 | 21.51 | 59.70 | 12.39 |
| √ | √ | weighted | **21.82** | 25.42 | **42.47** | **162.12** | 26.37 | 5.62 | **11.25** | **21.60** | **59.79** | **12.47** |

**Table 5: Ablation of vehicle ego-motion feature's effect on vehicle state estimation**

| Feature | Steer | | | | | | Velocity | | | | | |
|---|---|---|---|---|---|---|---|---|---|---|---|---|
| | $RMSE_{degree}$ | $A_{0.1}$ | $A_{0.5}$ | $A_{1.0}$ | $A_{5.0}$ | $A_{10.0}$ | $RMES_{m/s}$ | $A_{0.1}$ | $A_{0.5}$ | $A_{1.0}$ | $A_{5.0}$ | $A_{10.0}$ |
| R34 | 4.06 | 14.81 | 36.54 | 57.45 | 91.15 | 96.87 | 2.59 | 2.44 | 19.55 | 36.97 | 88.64 | 98.56 |
| Clip | 3.92 | 19.20 | 40.50 | 64.16 | 91.08 | 96.86 | 2.47 | 1.94 | 19.50 | 37.97 | 89.26 | 98.81 |
| I3D | 3.81 | **24.65** | **58.84** | 72.29 | 92.98 | 97.24 | **1.85** | **3.71** | 27.28 | **46.66** | 93.48 | 99.24 |
| $VEM_{r34}$ | **3.78** | 23.13 | 57.20 | **73.00** | **93.09** | **97.54** | **1.85** | 3.67 | **30.46** | 44.52 | **94.94** | **99.26** |

**Table 6: Ablation of vehicle ego-motion learning strategy's effect on event location and caption generation**

| Strategy | Event Location(avg) | | | Caption | | | | | | | | | |
|---|---|---|---|---|---|---|---|---|---|---|---|---|---|
| | | | | Description | | | | | Rationale | | | | |
| | Recall | Precision | F1 | B4 | M | R | C | S | B4 | M | R | C | S |
| R34+VSE | 50.08 | 58.95 | 53.76 | 19.01 | 23.30 | 39.16 | 146.68 | 24.35 | 4.68 | 9.94 | 18.96 | 50.36 | 11.04 |
| Clip+VSE | 50.66 | 60.27 | 55.64 | 19.53 | 23.74 | 41.10 | 157.49 | 25.01 | 5.21 | 11.00 | 19.65 | 55.65 | 12.20 |
| I3D+VSE | **51.38** | 61.84 | 56.13 | 21.08 | **25.82** | 42.20 | 160.61 | 25.63 | 5.28 | **11.58** | 19.95 | 58.08 | **12.88** |
| $VEM_{r34}$ + VSE | 50.79 | **63.05** | **56.26** | **21.82** | 25.42 | **42.47** | **162.12** | **26.37** | **5.62** | 11.25 | **21.60** | **59.79** | 12.47 |

5.28% for Recall/Precision/F1 respectively. Simultaneously, the caption task also achieves substantial enhancement. Despite the fact that rationale generation is still inferior to description, the improvements in rationale are much larger than Description. For instance, compared with Resnet34, the description sees an increase of about 14.78% with a rationale of 20.01%, demonstrating that our method indeed facilitates the model in mining explanatory information.

## 5 CONCLUSION

This paper presents the Ego Vehicle DVC, involving a multi-modal task with three sub-tasks targeting the investigation of dense video captioning in real-life first-person driving scenarios. Due to its distinctive observational viewpoint, this task comes with its unique challenges. We develop a strategy for learning vehicle ego-motion and a novel adjacent contrastive learning for boundary ambiguity. Extensive comparisons and ablation experiments demonstrate the effectiveness of our proposed method.

We suppose this topic is of great significance, whereas there are still some barriers hindering further research, a prominent problem is related to the dataset. The duration of the BDD-X dataset is relatively short, the descriptions and rationales lack diversity, and The recording frequency for vehicle states is too low. We may consider contributing a higher-quality dataset to this topic in the near future.

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
