# OpenReview forum: "Egocentric Vehicle Dense Video Captioning"
_acmmm.org/ACMMM/2024/Conference — MM2024 Oral_

### Official Review · Reviewer_CM5b · 2024-05-03

**Rating:** 5
**Confidence:** 3

**Summary:**

This paper proposes a novel dataset for dense video captioning on a vehicle's egocentric view, where not only event description but also rationale of the event are the target of mention. The authors also provide pre-training for ego-motion comprehension and a contrastive loss specially designed for this task, which contains three targets of mention. The experimental results show the challenge of this dataset and where we are currently at with this problem.

**Strengths:**

1. A dataset with unedited video footage is much more practical than one with edited web videos. This reviewer agrees that this dataset is novel and significant for the dense video captioning task.
1. The captioning target includes rationale, which enhances explainability of the decision and matches to autonomous driving applications.
1. The initial attempt of pre-training and adjacent contrastive learning seem reasonable and helpful for dataset followers.
1. The attempt achieves the best performance among baselines.

**Limitations:**

1. **[absence of Vid2Seq in the baseline]** Vid2Sec [66] (CVPR2023) is the State-of-the-art DVC method, but it is not in the baselines. The latest one in the current baselines [55] is an arXiv paper and is not authorized enough. This reviewer recommends testing Vid2Seq for better benchmarking of this new dataset.

2. **[Unclear vocabulary diversity in the dataset]** No statistics for the vocabulary diversity are provided. Such statistics are important to understand the difficulty of the task.

**Suitability:**

2

---

### Official Review · Reviewer_5A8t · 2024-05-20

**Rating:** 1
**Confidence:** 2

**Summary:**

The paper proposes a new task, Egocentric Vehicle Dense Video Captioning, aimed at understanding untrimmed, first-person driving videos through event location, event captioning, and vehicle state estimation. Unlike traditional dense video captioning, which focuses on third-person videos, this approach addresses the challenges of extracting relevant ego-motion information, describing driving behavior, and resolving boundary ambiguities. The method, validated on the BDD-X dataset, employs vehicle ego-motion learning and adjacent contrastive learning strategies, achieving new state-of-the-art performance on most metrics.

**Strengths:**

The paper's primary strength lies in applying dense video captioning (DVC) to first-person driving scenarios. This could be crucial for applications like development of semi-autonomous cars. This technology could be particularly relevant for companies focused on semi-self-driving vehicles, as it can help determine whether a driver or an AI agent is at fault in adverse situations.

**Limitations:**

1. Sorry to say, but the writing requires significant revision. There are numerous grammatical errors, and many sentences lack clear transitions and context.

    - The motivation in the abstract is not sufficiently relevant to highlight the importance of the work. The motivation feels lacking when transitioning from the first few sentences to the discussion on DVC on vehicles.

    - The paper lacks confidence in its claims, as seen in Lines 105-106 and in the conclusion sections.

    - Concepts like boundary ambiguity are not well positioned, despite their apparent importance to the authors. They require more context and explanation.

    - The discussion of vehicle state in terms of velocity seems trivial. More meaningful states would include being stopped, speeding, slowing down, hard braking, or sudden acceleration.

    - Although the paper hints that the ego-view is from the driver's perspective, it seems to be from the car's perspective. Including egocentric information from the driver's perspective would add more value to the work.

2. Lack of novelty in terms of algorithm

    - The paper does not introduce new algorithms; it is purely application-based.

3. Lacking motivations and explanations

    - Several scenarios in the paper lack sufficient motivation, such as the use of photometric reconstruction. The necessity of this approach is not explained.

    - The comparison between (a) and (b) in Figure 1 is misleading. The author's point about boundary ambiguity when comparing these figures is unclear.

    - The presented metrics, Recall and Precision, are insufficient to demonstrate the performance of this nature. Analyzing recall shows that the proposed model does not perform well. The author has not highlighted the importance of individual metrics.

The paper requires significant revision.

**Suitability:**

3

---

### Official Review · Reviewer_eaTE · 2024-05-20

**Rating:** 3
**Confidence:** 3

**Summary:**

1. The authors proposed a new task, Egocentric Vehicle Dense Video Captioning (Ego Vehicle DVC), including three sub-tasks: Event Location, Caption Generation, and Vehicle State Estimation. The dataset is adapted from BDD-X.

2. For the Ego Vehicle DVC task, an ego-motion information learning strategy and an adjacent contrastive learning strategy are introduced.

**Strengths:**

S1: The proposed Ego Vehicle DVC task is novel and challenging.

S2: Utilizing the existing ego-vehicle video dataset BDD-X for the Egocentric Vehicle Dense Video Captioning task is clever, and it saves a lot of data collection effort.

S3: The proposed method achieves SOTA results at most of the metrics.

**Limitations:**

W1: While leveraging the existing ego-vehicle video dataset is innovative, and the proposed method achieves state-of-the-art performance on it, the BDD-X dataset's quality is lacking. Issues such as short video durations and a lack of diversity impact the reliability of the method's performance.

W2: The proposed method exhibits a low recall on the Event Location task.

W3: The text contains too many typos.

**Suitability:**

3

---

### Official Review · Reviewer_yLnZ · 2024-06-03

**Rating:** 3
**Confidence:** 2

**Summary:**

In this paper, author propose a dense video captioning framework for first-person view driving videos. The problem statement is described and previous works in related fields are discussed. The method is evaluated on the BDD-X dataset and compared to various baselines on different sub-problems.

**Strengths:**

- The methodology and the model architecture are well described.
- The number of baselines chosen for comparison is sufficient. The experimental setting itself is extensively detailed.
- The paper has the merit of dealing with a partially unexplored problem.

**Limitations:**

- The fact that the problem is not well established makes comparison with existing works difficult.

- "However, the majority of DVC researchers focus on analyzing exocentric, or third-person videos, yet investigations into an egocentric, or first-person, viewpoint remain rare." This is a very strong assumption. There are some wide dataset of first-person videos used in active research (e.g. Ego4D). If this affirmation is relative to driving videos only, that should be better explicited.

- Formatting is somehow incosistent. References are sometimes spaced ("...two-stage methodologies [23 , 24 , 30..."), sometimes not ("...Learning from Bert[11]..."). Some subsection titles, for instance "Vehicle Ego-motion Extractor.", have a newline after, some other have not. This is not a methodological limitation of course but it hinders reading.

- Why is SSIM in Equation 3 multiplied by alpha/2 instead of just alpha? Even if this is a common practice and not proposed by the authors, a note explaining the reason behind the choice could be useful.

- Will you release the subset of BDD-X with additional video state data used for experiments? That's essential to prove reproducibility.

- Table 1 shows that the proposed method has low performance in terms of recall. Is there any task-specific reason to overlook recall performance? This is an important reason to discuss as the drop is consistent. More in general, a summary of method limitations should be presented in a dedicated paragraph.

I recognize the authors merits on dealing with an unconventional task. However, this brings to some non-trivial limitations in the evaluation which limits the impact of this work and make me doubt it is suitable for a top conference such as ACMMM, but I may be convinced otherwise.

**Suitability:**

3

---

### Meta-Review · Area_Chair_eFXZ · 2024-07-02

**Recommendation:** Accept (Oral)
**Confidence:** 5

**Metareview:**

After thoroughly checking reviews and rebuttal, I believe this study is worse to accept.